# MiR-21 Is Induced by Hypoxia and Down-Regulates *RHOB* in Prostate Cancer

**DOI:** 10.3390/cancers15041291

**Published:** 2023-02-17

**Authors:** Charlotte Zoe Angel, Mei Yu Cynthia Stafford, Christopher J. McNally, Heather Nesbitt, Declan J. McKenna

**Affiliations:** 1Genomic Medicine Research Group, Ulster University, Cromore Road, Coleraine BT52 1SA, UK; 2Patrick G Johnston Centre for Cancer Research, Queen’s University Belfast, Belfast BT9 7AE, UK

**Keywords:** prostate cancer, microRNA, miR-21, hypoxia, *RHOB*, biomarker

## Abstract

**Simple Summary:**

A low level of oxygen (hypoxia) is a common feature of many solid tumours. Tumour hypoxia is a contributing factor to prostate cancer progression and is known to cause the abnormal expression of many important genes, including microRNAs. In this study, we investigate the link between hypoxia and microRNA-21 (miR-21) in prostate cancer cells. We use in vitro and in vivo models to show that miR-21 expression is induced by hypoxia in prostate cells, which we propose explains why miR-21 up-regulation is a feature of prostate tumours. We demonstrate that miR-21 up-regulation can alter the behaviour of normal prostate cells and we further show for the first time in prostate cancer that it down-regulates *RHOB*, a tumour suppressor gene. We finish by presenting data to suggest miR-21 has considerable potential as a biomarker of hypoxia that can aid in the diagnosis and prognosis of prostate cancer.

**Abstract:**

Tumour hypoxia is a well-established contributor to prostate cancer progression and is also known to alter the expression of several microRNAs. The over-expression of microRNA-21 (miR-21) has been consistently linked with many cancers, but its role in the hypoxic prostate tumour environment has not been well studied. In this paper, the link between hypoxia and miR-21 in prostate cancer is investigated. A bioinformatic analysis of The Cancer Genome Atlas (TCGA) prostate biopsy datasets shows the up-regulation of miR-21 is significantly associated with prostate cancer and clinical markers of disease progression. This up-regulation of miR-21 expression was shown to be caused by hypoxia in the LNCaP prostate cancer cell line in vitro and in an in vivo prostate tumour xenograft model. A functional enrichment analysis also revealed a significant association of miR-21 and its target genes with processes related to cellular hypoxia. The over-expression of miR-21 increased the migration and colony-forming ability of RWPE-1 normal prostate cells. In vitro and in silico analyses demonstrated that miR-21 down-regulates the tumour suppressor gene Ras Homolog Family Member B (*RHOB*) in prostate cancer. Further a TCGA analysis illustrated that miR-21 can distinguish between different patient outcomes following therapy. This study presents evidence that hypoxia is a key contributor to the over-expression of miR-21 in prostate tumours, which can subsequently promote prostate cancer progression by suppressing *RHOB* expression. We propose that miR-21 has good potential as a clinically useful diagnostic and prognostic biomarker of hypoxia and prostate cancer.

## 1. Introduction

Tumour hypoxia, which refers to the development of poorly oxygenated regions in tumours due to chaotic growth patterns, is a well-established driver of poor outcomes in many solid tumours, including prostate cancer [1,2]. The cellular response to hypoxic stress involves a large network of overlapping signaling pathways and molecules which can promote tumour growth if they become dysregulated [2,3]. Prostate tumours are particularly hypoxic in comparison to normal prostate tissue, so it is very important to understand how this might contribute to the progression of this disease [1,4]. One aspect of the hypoxic response in prostate cancer which requires further research is the impact on microRNAs (miRNAs), short non-coding RNAs that negatively regulate target mRNAs and play an essential role in determining cell behaviour during hypoxia [5,6].

The abnormal expression of several miRNAs has been reported in prostate cancer, but the various ways in which they contribute to the development and progression of the disease remain to be fully explained [7,8]. More specifically, further research is needed to help understand how prostate tumour hypoxia can contribute to abnormal miRNA expression. This knowledge can then be used to improve the management of and treatment strategies for prostate cancer patients [9]. For example, miR-210 is a miRNA that has been consistently linked with hypoxia in various tissues [10]. We have previously reported research showing how it is up-regulated in prostate cancer and may contribute to prostate cancer development through its regulation of NCAM [11]. Others have shown a link between hypoxia and other miRNAs in prostate cancer cells, including miR-133a [12], miR-301a/b [13], miR-137 [14], miR-182 [15] and miR-145 [16]. However, there are still many miRNAs and targets that remain to be investigated in the context of prostate tumour hypoxia.

One such miRNA that has been linked to hypoxia in various cell types, as well as to being over-expressed in prostate cancer, is hsa-miR-21-5p (miR-21) [17]. In fact, miR-21 is one of the most studied miRNAs and it has been consistently shown to be over-expressed in many cancers, implying that it primarily acts as an oncogenic function [17,18]. Now considered a key ‘oncomiR’, miR-21 has been correlated with cancer incidence in numerous tissue types and has been repeatedly identified as a potential marker of advanced disease in various settings, including lung [19], colon [20], breast [21], cervical [22], pancreatic [23], liver [24] and oral [25] cancer. A large number of in vitro studies have also evidenced various functional roles and targets for miR-21 in these various cancer types (reviewed in [17,18,26]). Among these are studies proposing that hypoxia may be involved in causing miR-21 up-regulation in cancer cells. Key to this is that the miR-21 gene promoter has a binding site for hypoxia-inducible factor (HIF), the master orchestrator of the cellular hypoxic response [27]. It is therefore unsurprising that several studies have shown how hypoxia alters miR-21 expression in glioma [28], colon [29], pancreatic [30], lung [31], oral squamous cell carcinoma [32] and myeloma [33] cancer cells, among others. Additionally, there is also evidence from non-cancerous cells and tissues that miR-21 is involved in the hypoxic response during cardiovascular disease [34], pulmonary hypertension [35], renal ischemia/reperfusion injury [36] and erythropoiesis [37].

Given this substantial body of evidence, it seems reasonable to hypothesize that the hypoxia that occurs in prostate tumours would lead to miR-21 over-expression. Surprisingly, however, we are only aware of two studies to date that have specifically investigated this, both of which were limited to cell-line analyses. One study on DU145 prostate cancer cells showed a feedback mechanism between miR-21 and HIF-1α in regulating tumour angiogenesis, as well as demonstrating *PTEN* mRNA as a target [38]. The other showed that treating PC3 and LNCaP prostate cancer cells with a curcumin-derived synthetic analogue could decrease the hypoxia-induced expression of miR-21 but did not explore any mRNA targets [39]. Clearly more research is required, so in this paper we combine in vitro, in vivo and in silico approaches to investigate the link between hypoxia and the expression of miR-21 and selected targets in prostate cancer.

## 2. Materials and Methods

### 2.1. Cell Culture and Transfections

Cell-lines were obtained from American Type Culture Collection (ATCC, Rockville, MD, USA). Cells were frozen at low passage number and used within 6 passages after thawing. Cells were authenticated by an in-house genotyping service and routinely tested as mycoplasma-free (InvivoGen, Toulouse, France). Non-malignant prostate epithelial cell-line RWPE-1 was cultured in keratinocyte growth medium supplemented with 5 ng/mL human recombinant epidermal growth factor and 0.05 mg/mL bovine pituitary extract (Life Technologies, Paisley, UK). Human prostate cancer cell line LNCaP was cultured in RPMI-1640 supplemented with 10% FBS and L-glutamine (Life Technologies). All cells were grown in an incubator with a humidified atmosphere of 95% air and 5% CO_2_ at 37 °C and routinely passaged. For treatment in hypoxic conditions, cells were placed in normoxia (20% oxygen) or hypoxia (0.1% oxygen) at 37 °C in a hypoxia workstation (Ruskinn Technology, Bridgend, UK) for up to 72 h. For spheroid cell culture, 6-well plates were double-coated with a polyhema acrylamide layer (1.2 g polyhema (poly(2-hydroxyethyl methacrylate)) per 100 mL of 95% ethanol) to prevent cell adhesion to the base of the plate. A total of 30,000 LNCaP cells were seeded per well, media were replaced every 2–3 days and average spheroid size (*n* = 20) was measured by microscopy. For miRNA transfections, cells were seeded at appropriate density in a 6-well plate. Twenty-four hours after seeding, cells were transfected with miR-21 precursor (pre-miR-21) or non-targeting negative control precursor (pre-miR-neg) (both Life Technologies) at a final concentration of 25 nM using Lipofectamine 2000 according to manufacturer’s instructions (Life Technologies). After 48 h, cells were harvested for RNA extraction.

### 2.2. Colony Forming Assay

RWPE-1 cells were seeded as 500 cells per well of a 6-well plate, transfected after 24 h, and left for 12 days. The cells were rinsed with ice-cold phosphate buffered saline (PBS) and ice-cold fixing solution was gently added (7 parts methanol: 1 part acetic acid) for 5 min. Cells were stained with 0.5% crystal violet solution (Sigma, Poole, UK) in 85% methanol (15% deionized water) for 5 min and rinsed with cold water. To quantify the crystal violet, cells were lysed in 1% SDS solution in deionized water for 30 min at room temperature with rapid rocking. A total of 100 μL of the SDS solution was added to a well of a 96-well plate and the optical density was measured at 595 nm.

### 2.3. Migration Assay by Boyden Chamber

A total of 30,000 RWPE-1 cells were seeded into each well of xCELLigence^®^ CIM-16 plates and analysed using the RTCA DP Instrument (both ACEA Biosciences, San Diego, CA, USA). The upper chambers were filled with media lacking FBS/growth factors and the lower chamber with complete media to act as chemoattractant. The cells in the treatment and control conditions were normalized to serum-free media control wells (serum-free media in both the upper and lower chamber).

### 2.4. Quantitative Real-Time PCR (qRT-PCR)

Total RNA was extracted from cell lines, tumours and spheroids using Tripure^®^ reagent (Life Technologies). RNA integrity was confirmed by visualization on a 1% agarose gel (in tris-acetate-EDTA), and RNA concentration determined by NanoDrop™ 2000 spectrophotometer (ThermoFisher Scientific, Horsham, UK). A total of 1µg RNA was used for first strand cDNA synthesis using random primers with Transcriptor high-fidelity cDNA synthesis kit (Roche, Sussex, UK) according to manufacturer’s instructions. Amplification of PCR products was quantified using FastStart SYBR Green Master (Roche) on a Roche LC480 Lightcycler, using primer sets for *RHOB* (fw: CGACGTCATTCTCATGTGCT, rv: CGAGGTAGTCGTAGGCTTGG) *PTEN* (fw: ACCCACCACAGCTAGAACTT, rv: GGGAATAGTTACTCCCTTTTTGTC), *HPRT* (fw: CCTGGCGTCGTGATTAGTGA, rv: CGAGCAAGACGTTCAGTCCT). Expression was normalized to *HPRT* and graphs represent the combined results of three independent biological replicates.

qRT-PCR of miRNAs was performed using the miRCURY LNA^TM^ microRNA PCR system (Qiagen, Manchester, UK). A total of 20 ng template RNA was used in each first strand cDNA synthesis reaction. PCR was performed with over 40 amplification cycles and fluorescence was monitored on a LC480 Lightcycler (Roche). Normalization was against U6snRNA or SNORD48. Serum RNA was extracted from whole blood using the miRNeasy Serum/Plasma Kit (Qiagen) using 5 μL serum. miR-191 was used as housekeeping control for PCR analysis of serum miRNA expression. For all qRT-PCR miRNA analyses, graphs represent the combined results from 3 independent biological replicates, unless otherwise indicated.

### 2.5. Protein Analysis

Protein was extracted using urea buffer. Western blots were performed using the Invitrogen NuPAGE^®^ Novex^®^ Gel System and reagents (ThermoFisher Scientific). Antibodies used for blotting were anti-RhoB (Proteintech, Manchester, UK) and anti-HIF-1α (Sigma), with anti-α-Actin (Sigma) or anti-GAPDH (Proteintech) as loading control, with overnight rocking at 4 °C. Membranes were blocked in tris-buffered saline with 5% Marvel (Premier Foods, Hertfordshire, UK) and 0.05% Tween 20 (Sigma) for 1 h at room temperature followed by incubation in the appropriate secondary antibody (goat anti-rabbit IgG-HRP or goat anti-mouse IgG-HRP (1:10,000) (both Santa Cruz Biotechnology, Dallas, TX, USA) for 1 h at room temperature. Luminescence was revealed by incubation with enhanced chemiluminescent reagent (Life Technologies) and signal detected on a G:BOX F3 imaging system (Syngene, Cambridge, UK).

### 2.6. In Vivo Experiments

All experimental procedures were carried out in accordance with the Animal (Scientific Procedures) Act 1986 and the UKCCCR guidelines for the welfare of animals in experimental neoplasia [40]. Animal studies are reported in compliance with the ARRIVE guidelines [41]. All animal work was carried out in the Biomedical and Behavioural Research Unit at Ulster University under the establishment license (No. 5007) and project license (PPL2808), both granted and approved by the Department of Health, Northern Ireland. Ethical approval for PPL2808 was sought and obtained from the Animal Welfare and Ethical Review Body at Ulster University. Eight–ten-week-old male nude mice weighing 25–30 g (Envigo, Indianapolis, IN, USA) were housed under standard laboratory conditions in a temperature-controlled (22 °C; 50–55% humidity) specific pathogen-free environment with a 12-h light/dark cycle. Food and water were supplied ad libitum. Procedures and administrations were performed using aseptic technique, and tumour implantation and oxygen electrode measurement were performed under anaesthesia. For xenograft establishment, mice were briefly anaesthetized by inhalation of isoflurane. LNCaP xenografts were established by subcutaneous injection of 5 × 10^6^ cells suspended in 100 μL of ice-cold matrigel (growth factor reduced) (Corning, Flintshire, UK) to the dorsum, using an ice-cold 21 g needle. Once the tumour became palpable, dimensions were measured using Vernier calipers, using the formula: volume = (height × height × width)/2. Oxygen electrode measurement was performed using an OxyLite^®^ fibre optic probe (Oxford Optronix, Abingdon, UK) which was inserted into the tumour through a 21 g needle. After the probe readings had normalized, 30 readings were recorded per site (the median reading was used). At least two sites (with similar readings) were measured per tumour and the mean of the 2 median readings was taken to represent the tumour. For drug administration, bicalutamide (Sigma) was prepared in vehicle (0.1% DMSO in corn oil) and administered orally via gavage at 6 mg/kg daily. When tumour volume reached 150 mm^3^, mice were randomly assigned to treatment groups and dosing was initiated. Mice were sacrificed by cervical dislocation and tumours were excised immediately using aseptic technique. Experimental endpoints were established as tumours reaching gross mean diameter of 12 mm, or loss of >15% of animal body weight.

### 2.7. Databases

To identify mRNA targets of the miRNAs, miRTarBase (http://mirtarbase.cuhk.edu.cn/, accessed on 10 March 2019) [42] was searched. The Cancer Genome Atlas Prostate Adenocarcinoma (TCGA-PRAD) repository data were accessed at http://portal.gdc.cancer.gov/projects (accessed on 11 January 2022). Analysis of pre-processed, normalized TCGA-PRAD data was performed using The University of California Santa Cruz UCSC’s Xena Functional Genomics Explorer (UCSC Xena) (http://xenabrowser.net/, accessed on 13 February 2022) [43], CancerMIRNome (http://bioinfo.jialab-ucr.org/CancerMIRNome/, accessed on 21 February 2022) [44] and Firebrowse (http://firebrowse.org/, accessed on 11 May 2019) analysis tools. Regulome Explorer (http://explorer.cancerregulome.org/, accessed on 13 December 2019) was used to analyse a primary prostate cancer dataset from a single study [45]. Serum miR-21 expression data were analysed from prostate cancer samples selected from the Gene Expression Omnibus (GEO) (http://www.ncbi.nlm.nih.gov/geo/, accessed on 13 May 2022) datasets GSE112264 [46], GSE139031 [47], GSE113486 [48] and GSE134266 [49]. Additional survival analysis was performed using Kaplan–Meier plotter (KM-Plotter) (http://kmplot.com/analysis/, accessed on 31 May 2022) [50]. Functional enrichment analysis on TCGA-PRAD data was performed using clusterProfiler, a Bioconductor package for gene classification and enrichment analyses, based on statistical analysis of Kyoto Encyclopedia of Genes and Genomes (KEGG) and biomedical gene set databases [51,52]. This tool is integrated into CancerMIRNome, which is specifically designed to analyse miRNA expression in TCGA samples, following data processing and normalization using the bioinformatics pipeline in R/Bioconductor package GDCRNATools [53]. Network analyses were performed and visualized using GeneMANIA (http://genemania.org/, accessed on 3 June 2022) [54] and miRTargetLink 2.0 (http://ccb-compute.cs.uni-saarland.de/mirtargetlink2, accessed on 4 June 2022) [55].

### 2.8. Statistics

All in vitro experiments were performed in triplicate (three independent measures). Graphs were generated using Graphpad PRISM v6. All bar graphs show mean ± standard error of at least three biological replicates (independent measures), with statistical significance assessed by paired *t*-test. All boxplots and scatterplots were based on data downloaded from the databases above. Boxplots show mean and Tukey whiskers with statistical significance assessed by either unpaired *t*-test with Welch’s correction or non-parametric Kruskal–Wallis one-way ANOVA with Dunn’s multiple comparison test. Statistical significance for scatterplots was assessed by Pearson’s correlation with *p*-values adjusted for multiple hypothesis testing. For Firebrowse analysis, Spearman’s rank correlation and two-tailed *p*-values were estimated using ‘cor.test’ function in R. Statistical significance for Kaplan–Meier graphs was assessed by log-rank (Mantel-Cox) test. The random forest model was prepared using RStudio, based upon TCGA-PRAD data for miR-21, miR-210, Gleason grade and remission. Model based on 500 decision tree iterations to generate accuracy, specificity, sensitivity and AUC data. Statistical significance was assessed by McNemar’s test. For multiple hypothesis correction, the adjusted *p*-value (*Q*-value/false discovery rate (FDR)) used Benjamini and Hochberg procedure which was applied in clusterProfiler package. For all figures, data were considered significant when * *p* < 0.05, ** *p* < 0.01, *** *p* < 0.001.

## 3. Results

### 3.1. Up-Regulation of miR-21 Is Associated with Prostate Cancer

We first performed an analysis of the TCGA-PRAD patient cohort to confirm that the expression of miR-21 was significantly up-regulated in prostate cancer tumour tissue compared to that of normal prostate tissue (Figure 1A). Likewise, using separate GEO datasets, we also showed that the serum expression of miR-21 was significantly increased in the prostate cancer patients compared to that of the non-cancerous control patients (Figure 1B) and in the benign prostatic hyperplasia (BPH) patients compared to the healthy controls (Appendix A). A further UCSC Xena analysis of the TCGA-PRAD data revealed that a higher miR-21 expression was significantly associated with clinicopathological markers of prostate cancer progression, including Gleason score, pathological T stage and lymph node involvement (Figure 1C–E). This significant association was confirmed using a separate Firebrowse analysis tool (Figure 1F).

A functional enrichment analysis further revealed that miR-21, by virtue of targeting many cancer-related genes, was significantly associated with several gene set description terms related to prostate cancer (Table 1). Together, this data clearly demonstrated that the up-regulation of miR-21 expression plays a significant biological role in prostate cancer development and progression.

### 3.2. miR-21 Is Up-Regulated by Hypoxia in Prostate Cells

We hypothesized that the up-regulation of miR-21 observed in prostate cancer was caused by hypoxia, which is a common feature of prostate tumours. To explore this, we utilized three models of hypoxia to measure the expression of miR-21. Using a hypoxic chamber, LNCaP cells were cultured at 0.1% oxygen (hypoxia) or 20% oxygen (normoxia) for 24 h or 48 h, since prostate tumours are typically very hypoxic and estimated to have oxygen percentages as low as 0.1% [1]. In parallel with this, we also cultured LNCaP spheroids, which we knew from previous studies develop hypoxia as they increase in size [11]. In both models, we observed that cellular hypoxia significantly increased miR-21 expression (Figure 2A,B). Hypoxia in both LNCaP cell models was confirmed by observing increased levels of HIF-1*α* (Appendix A), as previously reported [11].

We validated these observations in vivo, using a xenograft LNCaP tumour model. We had previously demonstrated that treatment with bicalutamide in this model consistently induced hypoxia by causing the collapse of tumour vasculature [56,57]. As expected, this hypoxic stress resulted in the increased expression of miR-21 at Day 7 and Day 28 of bicalutamide in both tumour and serum, although variation in animal data resulted in non-significant *p*-values (Figure 2C,D). We also wanted to compare miR-21 with miR-210, since our previous work proved that miR-210, a well-established marker of hypoxia, was significantly increased in hypoxic prostate cancer cells. There was a highly significant positive correlation between the two miRNAs in the TCGA-PRAD biopsy data corroborating a role for miR-21 in the hypoxic response in prostate cancer (Figure 2E). Finally, we carried out a functional enrichment analysis to reveal a highly significant association for miR-21 and its target genes with biological processes related to cell hypoxia, oxygen levels and oxidative stress (Table 2). We conclude that the combination of these various analyses provides strong evidence that the associated tumour hypoxia is likely to be a contributing factor to the up-regulation of miR-21 in prostate cancer.

### 3.3. Ras Homolog Family Member B (RHOB) Is Down-Regulated by miR-21 in Prostate Cancer

We were interested in identifying a target of miR-21 which had not previously been demonstrated in prostate cancer. Given that miR-21 appears to have an oncogenic role in this setting, we sought to identify a target which might play a tumour suppressor role, the expression of which would be reduced by the up-regulated miR-21 levels. Among the targets listed in Table 2, we noted Ras Homolog Family Member B (*RHOB*) as an interesting candidate. This gene codes for RhoB, a Rho family GTPase that has been attributed a tumour suppressor role in many cancers [58]. RhoB has also been linked to hypoxia [59,60] and has been validated as a target of miR-21 in various cell types [61,62,63], but no study to date has explored the link between miR-21 and *RHOB* expression in prostate cancer. A PCR confirmed that the over-expression of miR-21 in LNCaP cells using a transient transfection of the pre-miR-21 precursor molecule resulted in a significant reduction in *RHOB* mRNA levels, as well as *PTEN*, an established target of miR-21, relative to the scrambled control (Figure 3A). A Western blot confirmed that the over-expression of miR-21 reduced the RhoB protein levels in LNCaP cells (Figure 3B). In the same bicalutamide-treated LNCaP tumours in which we observed miR-21 up-regulation, the *RHOB* mRNA levels were significantly reduced by Day 28, relative to those of the vehicle-treated tumours (Figure 3C).

We hypothesized that increased hypoxia in prostate epithelial cells and tumours induces miR-21 which in turn down-regulates *RHOB*; thus, we explored TCGA-PRAD datasets to investigate if miR-21 and *RHOB* expression shows a reciprocal expression pattern in prostate tissue. We found that a significant inverse correlation does indeed exist between miR-21 and *RHOB* gene expression in prostate tissue, as we would expect if *RHOB* was a target of miR-21 (Figure 3D). A UCSC Xena analysis of TCGA-PRAD data revealed that *RHOB* was significantly down-regulated in prostate cancer tumour tissue compared to that of normal prostate tissue (Figure 3E). Furthermore, lower *RHOB* expression was significantly associated with Gleason score and stage (Figure 3F,G). This was confirmed using a separate Firebrowse analysis tool (Figure 3H). This *RHOB* expression profiling consistently showed the opposite trend to that observed for miR-21 (shown in Figure 1), implying that it is targeted by miR-21 in prostate cancer.

### 3.4. miR-21 Over-Expression Increases Migration and Colony-Forming Ability of RWPE-1 Cells

The above data present a potential biological mechanism whereby the increased expression of miR-21 down-regulates *RHOB*, which may drive tumour advancement. To explore this further, we looked at the effect of over-expressing miR-21 in RWPE-1 cells, an immortalized but non-cancerous prostate cell line. We found that the overexpression of miR-21 resulted in a highly significant increase in cell migration (Figure 4A). Moreover, the ability of RWPE-1 cells to form colonies was significantly increased following miR-21 overexpression (Figure 4B,C). We also confirmed that the miR-21 over-expression significantly decreased the *RHOB* mRNA levels in these cells (Figure 4D). A network analysis of the *RHOB* interactions demonstrates how altered expression caused by increased miR-21 levels subsequently impacts many other genes and proteins which play important roles in RhoB-regulated processes (Appendix A). This provides evidence for how miR-21 over-expression can drive prostate cells toward a more cancerous phenotype. It is also worth remembering that miR-21 up-regulation also impacts cell behaviour through the rest of its regulatory network, which includes many genes strongly linked with prostate cancer (Appendix A).

### 3.5. Potential of miR-21 as a Biomarker of Prostate Cancer

Given the consistent association of miR-21 with various prostate cancer clinical parameters, there is good potential for miR-21 as a diagnostic and/or prognostic biomarker in this setting. The ROC curve analysis of the TCGA-PRAD cohort demonstrates that miR-21 shows high potential for distinguishing between tumour and normal tissue (Figure 5A). Similarly, biochemical recurrence following therapy is associated with significantly high levels of miR-21 (Figure 5B) and there is significant difference in the miR-21 levels between groups of patients who show different remission responses after primary therapy (Figure 5C). The Kaplan–Meier graphs show high levels of miR-21 are significantly associated with decreased overall survival and reduced disease-free interval (Figure 5D,E), even though the number of deaths in this cohort is low. We also performed similar analyses for *RHOB* and showed that biochemical recurrence following therapy is associated with significantly low levels of *RHOB*, although the Kaplan–Meier graphs showed no significant association with the overall survival or disease-free interval (Appendix A). With a view to clinical application, we think the proxy marker(s) for hypoxia might be useful for monitoring prostate tumour growth and response to therapy. For example, combining miR-210 and miR-21 measurements with one’s Gleason score could help predict remission following therapy. As a representative illustration of this multi-variate approach, a random forest mathematical model based on 500 decision tree iterations of these three variables had a >90% accuracy, >50% sensitivity and 100% specificity in predicting treatment outcomes in the TCGA-PRAD patient cohort (Appendix A). Combining miR-21 with other measurements may therefore be the key to stratifying patients effectively into risk categories during prostate cancer management. Furthermore, since miR-21 over-expression has been consistently linked with many other cancers, it was no surprise to find that high miR-21 expression in tumour tissue is associated with significantly poorer survival in several other TCGA patient cohorts, indicating that it could be a useful biomarker for different cancers (Appendix A).

## 4. Discussion

Although miR-21 has been studied in several cancers and is generally accepted to have an oncogenic role, the effect of prostate cancer hypoxia on miR-21 has not been well studied, so we wanted to explore that relationship further in this study. This is the first report to present research showing how the hypoxia-induced up-regulation of miR-21 can contribute to prostate cancer development through its regulation of *RHOB*.

We first established through various analyses of TCGA and GEO prostate cancer datasets that miR-21 over-expression is indeed associated with prostate cancer and the progression of the disease (Figure 1, Table 1), in accordance with previous findings [64,65]. We then demonstrated in three LNCaP models of prostate cancer hypoxia that miR-21 levels were increased in response to hypoxia, and also demonstrated a close association with hypoxia-related cellular mechanisms through its network of target genes (Figure 2, Table 2). From this data, we conclude that hypoxia is likely to be a key contributor to the up-regulation of miR-21 in prostate cancer cells, which is detectable as increases in the serum levels of miR-21, either due to an active export mechanism or the “spillover” from necrotic tumour cells. This not only validates in vitro observations previously found in prostate cancer cells [38,39], but also adds further in vivo and in silico evidence demonstrating the relevance of this to the hypoxic prostate tumour environment. Additionally, the data corroborate findings from other cancers which have established miR-21 as a hypoxia-induced miRNA [28,29,30,31,32,33]. It is also worth highlighting that miR-21 can be profiled in serum samples from humans (Figure 1B and Appendix A) and animals (Figure 2D), suggesting it has value as a circulating biomarker of hypoxia and/or prostate cancer. Interestingly, miR-21 serum levels were also elevated in benign prostatic hyperplasia (BPH) patients compared to healthy controls (Appendix A). Tissue ischemia is also a feature of BPH [66], so it is feasible that hypoxic cells in this condition up-regulate miR-21. Indeed, a recent study has shown that miR-21 levels are elevated in urine from BPH patients, compared to that of healthy controls [67]. BPH is also associated with tissue infarction which could explain why serum miRNA levels may reflect prostatic tissue levels, as necrotic cells leach their contents into the blood.

Measuring the circulating levels of miR-21 in serum or urine would be less invasive than tissue biopsy profiling and would also allow for longitudinal bio-fluid sampling to monitor disease progression or response to treatment. Although many studies have successfully measured circulating miR-21 levels in humans and animals, very few have specifically implicated this as a marker of hypoxia. Our data suggest that the release of miR-21 from prostate cancer cells into serum (or urine) make it an attractive candidate for monitoring changes in tumour hypoxia status, especially since it is present in relative abundance and therefore easily detectable. Indeed, studies on hypoxia in liver cancer [68,69] and glioma [70] have shown how circulating levels of miR-21 and other miRNAs can track patients’ response to treatment. Moreover, circulating levels of miR-21 have been successfully measured in non-cancerous human and animal studies of hypoxia, including diabetic rat models [71], pulmonary hypertension patients [72], Crohn’s disease sufferers [73] and in maternal screening for fetal hypoxia during pregnancy [74]. Biologically, the cellular release of miR-21 from cells is important as there is also a likely systemic effect on neighbouring cells. For example, studies on various tissue types have reported that hypoxic cells release exosomes with elevated miR-21 levels, which then impact the behaviour of other cells, promoting proliferation, migration, invasion, angiogenesis and the epithelial-to-mesenchymal transition (reviewed in [75]). Unsurprisingly, exosomal miR-21 has been shown to promote the growth of many cancer cell types, including renal [76], ovarian [77], pancreatic [78] and liver [79]. Taken together, these findings indicate that the hypoxia-induced release of exosomal miR-21 could be an important mediator of paracrine signaling in the prostate tumour microenvironment, thereby promoting cancer progression.

To explore possible biological pathways involved in this, we then wanted to identify a potential target of miR-21 that had not previously been shown in prostate cancer cells. From Table 2, we identified *RHOB* as a candidate of interest. *RHOB* codes for the tumour suppressor RhoB and has been validated as a target of miR-21 in breast [61], colorectal [62] and cervical [63] cancer cells, but this is the first study to link miR-21 and *RHOB* in prostate cancer. In this study, we demonstrate that miR-21 over-expression significantly reduces *RHOB* mRNA levels in prostate cells in vitro and in vivo (Figure 3A,C and Figure 4E). We also confirmed that miR-21 down-regulated RhoB at the protein level in LNCaP cells which is further proof that it is a target (Figure 3B). A further analysis of the TCGA data showed a significant inverse correlation between miR-21 and *RHOB* (Figure 3). This relationship with clinicopathological measurements was the opposite of the profile seen for miR-21 (Figure 1), as expected if *RHOB* was a target.

The regulation of RhoB by miR-21 is significant because activated Rho GTPases, such as RhoB, bind to a variety of downstream effectors that impact cell migration, invasion, cell division, wound healing and actin reorganization, among other cellular processes [58]. In relation to this study, RhoB was also interesting because it had been linked to hypoxia-induced responses in different cell types, including inflammation, apoptosis and migration [59,60,80]. However, this is the first report to implicate RhoB in the cellular response to hypoxia in prostate cancer. Since RhoB is largely understood to play a tumour-suppressor role in this disease, its down-regulation by elevated miR-21 levels would be detrimental, increasing the propensity for carcinogenic growth. Indeed, the depletion of RhoB in prostate cancer cells has been shown to reduce cell–cell adhesion [81], inhibit apoptosis [82] and increase cell migration [83]. In this paper, we add to this body of evidence by showing that miR-21 decreases *RHOB* expression in non-cancerous RWPE-1 prostate cells, concomitant with the significantly increased migration and colony-forming ability of these cells (Figure 4). It is also worth noting that studies have shown how the specific localization of RhoB to endosomes permits it to regulate the trafficking and recycling of numerous proteins, such as the oncogenic intracellular kinases Src and Akt, and the oncogenic receptors EGFR and CXCR2, which have been co-localized with RhoB in endosomes of varying stages [58]. Therefore, insufficient levels of RhoB could result in the accumulation of these oncogenes. Taken as a whole, these data indicate that the down-regulation of RhoB by miR-21 could have implications for both prostate cancer cell function and the regulation of the tumour microenvironment as a whole. This is highlighted by our findings showing that low *RHOB* expression is significantly associated with prostate cancer and clinical markers of disease progression (Figure 3). Although in vivo experiments involving the over-expression and/or knockdown of either molecule in mice were beyond the scope of this study, this would be a sensible approach in any future work looking to unravel the biological mechanisms that link miR-21 and RhoB in prostate cancer. There are few data from in vivo experiments investigating hypoxia and miR-21 in prostate cancer, but others have had success using murine models to investigate hypoxia-related miR-21 effects in glioma [84] and lung cancer [85].

As mentioned above, miR-21 has been the subject of much research assessing its potential use as a clinical biomarker, with systematic reviews concluding that it has considerable value as a diagnostic and prognostic marker in cancer [86,87,88,89]. Similarly, a recent systematic review and meta-analyses from our research group concluded that elevated levels of miR-21 are associated with poor prognoses in prostate cancer patients, but acknowledged that better-designed, standardized studies are required for it to gain clinical acceptance as a robust prognostic biomarker for this disease [90]. Nevertheless, with continued research, it would seem likely that miRNAs will gain acceptance as biomarkers for cancer, with particular emphasis on their utility as non-invasive and/or liquid biopsy measurements [91,92]. If so, we propose that miR-21 would be a leading candidate to profile in prostate cancer, given its obvious biological importance in prostate cancer progression and relatively high expression levels in tissue, serum and urine [18,93,94]. Specifically, we suggest that using miR-21 to track the hypoxic status of prostate tumours could be very valuable, since our previous research has shown how the bicalutamide-induced changes in tumour vasculature and hypoxia can select for more aggressive cancer cells. This helps explain why some patients relapse after therapy, so the ability to monitor tumour hypoxia post-treatment would be useful. Indeed, a recent meta-analysis study identified miR-21 as one of several miRNAs that could predict responses to androgen-deprivation therapy [95]. Elsewhere, various research groups have developed hypoxic signature panels of gene markers to predict prostate cancer prognoses and treatment outcomes, although notably these did not include microRNAs [96,97,98,99]. Our research suggests that hypoxia-induced miRNAs should really be included in such panels, especially if blood-based markers are preferred. Interestingly, *RHOB* was included in the hypoxic signature panel for predicting systemic metastasis [98]. However, a major challenge to its successful clinical application is the inherent heterogeneity of prostate tumours, which complicates their accurate diagnosis and treatment [100,101]. It is important to realize that the overall tumour expression of miR-21 and *RHOB* will be determined by a variety of cell types, such as immune cells and fibroblasts, as well as tumour cells. Likewise, there may be multiple tumour foci present, representing tumour cells with distinctly different molecular characteristics, which provides significant challenges for modelling the disease in experiments [102]. Future work to address this is likely to require single-cell analyses coupled with advanced proteomics to gain further insight into the intra-tumour and inter-tumour heterogeneity present in prostate cancer. These approaches would help in identifying the precise pattern of their genomic and/or proteomic expression of miR-21 and *RHOB*, to help unravel their combined biological function.

Even with this sort of focused analysis, however, it is expected that miRNA measurements would have to be added to more traditional markers to help improve their diagnostic or prognostic accuracy. We have previously demonstrated that multivariate panels are much more likely to have diagnostic and prognostic value in PCa than single biomarkers [103,104]. With that in mind, we prepared a representative statistical model in this study, proposing how miR-21, miR-210 and Gleason scores can be combined to predict patient outcomes in the TCGA-PRAD dataset with >90% accuracy (Appendix A). We acknowledge that such a model would need to be properly validated with independent training and test data, but this illustrates how the measurement of hypoxia-related miRNAs could be practically used as a prognostic indicator. Indeed, current models for prostate cancer risk prediction utilize various combinations of genomic, proteomic and/or clinical measurements, including the Stockholm-3 risk-based model [105], the 4kscore [106] and the European Randomised Study of Screening for Prostate Cancer risk calculator [107]. We propose that miR-21 could be a useful addition to the list of variables to be included, either as a tissue- or bio-fluid-based marker, as these models evolve to better inform evidence-based decision making for the clinical management of PCa patients. Naturally, miR-21 may also be a useful biomarker for other cancers, as we have highlighted (Appendix A).

## 5. Conclusions

This is the first study to show that the hypoxia-induced expression of miR-21 in prostate cells can promote prostate cancer progression by down-regulating the tumour suppressor gene *RHOB*. We have shown that miR-21 is significantly up-regulated by hypoxia in prostate cells in vitro and in vivo, as well as demonstrated that high levels of miR-21 expression are associated with prostate cancer and clinicopathological markers of disease progression. This study also highlights the potential of miR-21 as a diagnostic or prognostic marker in prostate cancer, in combination with other markers, such as miR-210.

## Figures and Tables

**Figure 1 cancers-15-01291-f001:**
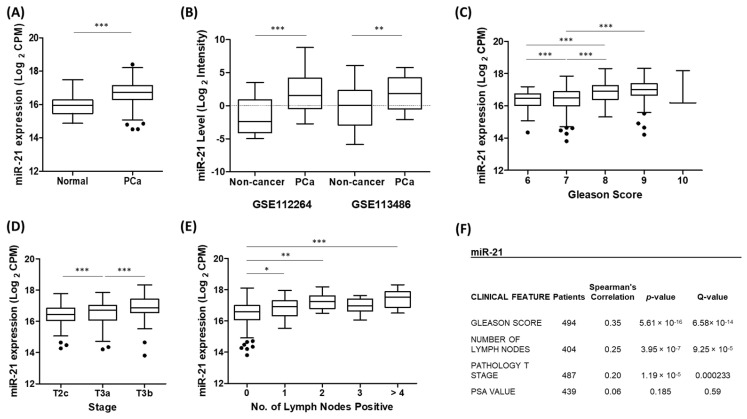
Up-regulation of miR-21 is associated with prostate cancer. (**A**) UCSC Xena analysis of TCGA-PRAD samples shows miR-21 expression is significantly increased in prostate tumour tissue (*n* = 494) compared to that of normal prostate tissue (*n* = 52). (Welch’s *t*-test *** *p* < 0.001). (**B**) miR-21 is significantly elevated in the serum of prostate cancer patients compared to that of healthy, non-cancer control patients. Data are from GEO datasets GSE112264 (*n*, Non-cancer = 41, PCa = 809) and GSE113486 (*n*, Non-cancer = 100, PCa = 40). (One-way ANOVA with multiple comparison tests, ** *p* < 0.01, *** *p* < 0.001.) UCSC Xena analysis of TCGA-PRAD samples shows expression of miR-21 is significantly associated with (**C**) Gleason grade (**D**) pathological T stage and (**E**) number of positive lymph nodes. (One-way ANOVA with multiple comparison tests, * *p* < 0.05, ** *p* < 0.01, *** *p* < 0.001). (**F**) Firebrowse analyses of TCGA-PRAD data (*n* > 400) confirm miR-21 expression had significant positive correlation with Gleason score, number of positive lymph nodes and pathological T stage. (*p*- and *Q*-values were generated by Spearman’s correlation with multiple hypothesis correction.) All boxplots show mean and Tukey whiskers.

**Figure 2 cancers-15-01291-f002:**
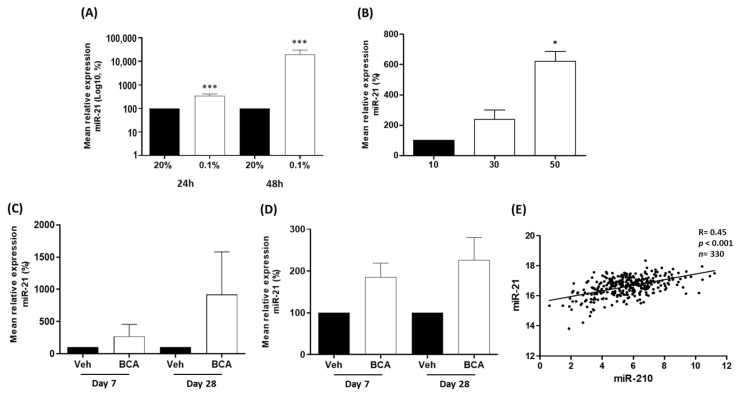
miR-21 is up-regulated by hypoxia in vitro and in vivo. (**A**) After 24 and 48 h (h) of hypoxia (0.1% oxygen), miR-21 expression is significantly increased relative to atmospheric oxygen levels (20%) in LNCaP cells. (**B**) In LNCaP spheroids, miR-21 expression is significantly elevated as spheroid size (and associated hypoxia) increases. (Both paired *t*-test, * *p *< 0.05, **** p *< 0.001). In bicalutamide-treated (BCA) mice with xenograft LNCaP tumours, miR-21 is increased in the (**C**) tumours and (**D**) serum at Day 7 and Day 28, relative to vehicle-treated (Veh) animals (*n* = 4 mice per group). (**E**) Regulome Explorer analysis revealed significant positive correlation between expression of hypoxia-induced miR-210 and miR-21 (Pearson correlation, *p *< 0.001). All bar graphs show mean ± SEM of at least three biological replicates.

**Figure 3 cancers-15-01291-f003:**
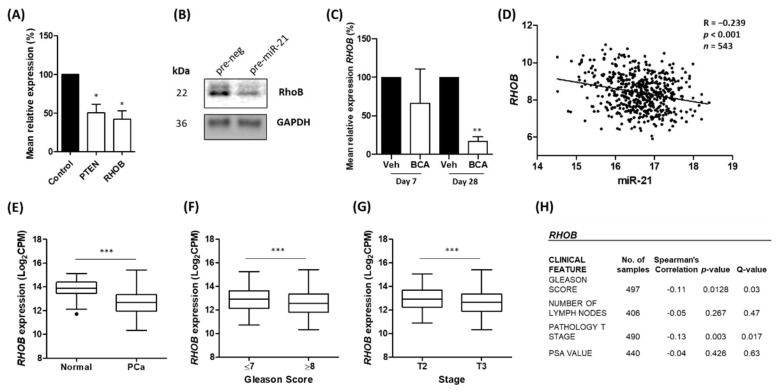
*RHOB* expression is inversely correlated with miR-21 in prostate cancer. (**A**) Overexpression of miR-21 results in significant reduction in the levels of *RHOB* mRNA levels in LNCaP cells, as well as reduction in the levels of *PTEN*, a well-established miR-21 target. (Paired *t*-test, * *p* < 0.05). (**B**) Representative Western blot shows over-expression of miR-21 causes down-regulation of RhoB protein in LNCaP cells (Appendix A). (**C**) In bicalutamide-treated LNCaP tumours (BCA), *RHOB* is significantly reduced by Day 28 relative to that of vehicle-treated tumours (Veh). (Paired *t*-test, *** p* < 0.01). (**D**) CancerMIRNome analysis of TCGA-PRAD samples, including normal (*n* = 52) and tumour (*n* = 491) tissue samples, shows the expressions of miR-21 and *RHOB* are significantly negatively correlated (Pearson correlation, *p* < 0.001). UCSC Xena analysis of TCGA-PRAD samples shows (**E**) *RHOB* expression is significantly reduced in tumour (*n *= 498) tissues relative to normal (*n* = 52) tissue and significantly decreases with (**F**) Gleason score and (**G**) pathological T stage. (All Welch’s *t*-test, **** p* < 0.001.) (**H**) Firebrowse analyses of TCGA-PRAD data (*n* > 400) confirm *RHOB* expression has significant negative correlation with Gleason score, number of positive lymph nodes and pathological T stage. (*p*- and *Q*-values were generated by Spearman’s correlation with multiple hypothesis correction.) All bar graphs show mean ± SEM of three biological replicates. All boxplots show mean and Tukey whiskers.

**Figure 4 cancers-15-01291-f004:**
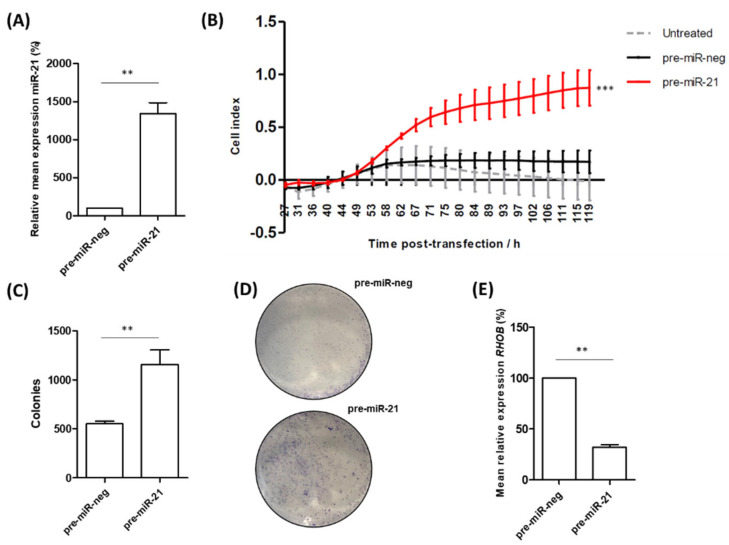
miR-21 over-expression increases migration and colony-forming ability of RWPE-1 cells. (**A**) Confirmed over-expression of miR-21 in RWPE-1 cells resulted in (**B**) significantly increased capability for migration of the pre-miR-21 transfected cells, relative to untreated and the scrambled negative control (pre-miR-neg). (One-way ANOVA, **** p* < 0.001.) Mean ± SEM of three biological replicates is shown. (**C**) Cells transfected with miR-21 have significantly higher colony-forming ability compared to cells transfected with pre-miR-neg as shown by quantified colony forming assays (*n* = 3) and (**D**) representative images of crystal violet colony staining. (**E**) Confirmation that *RHOB* mRNA levels are reduced in RWPE-1 cells following overexpression of pre-miR-21. (All bar charts; paired *t*-test, *** p* < 0.01.)

**Figure 5 cancers-15-01291-f005:**
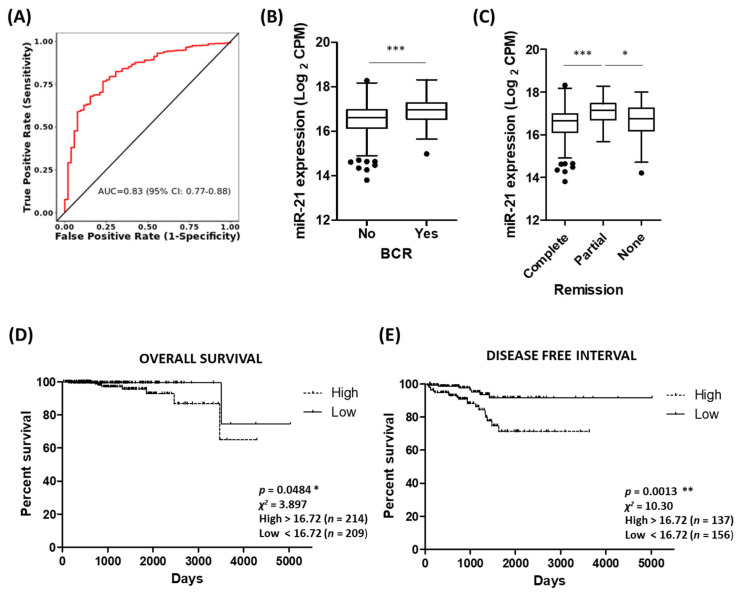
Potential of miR-21 as a biomarker of prostate cancer. (**A**) ROC curve analysis demonstrating that miR-21 shows high potential for distinguishing between tumour and normal tissue. Analysis performed using CancerMIRNome based on PRAD cohort in TCGA database. (**B**) Biochemical recurrence is associated with significantly high levels of miR-21 (*n*, no recurrence = 406, recurrence = 61). (Welch’s *t*-test, *** *p* < 0.001.) (**C**) Significant difference in miR-21 levels between patient remission response after primary therapy (*n*, complete = 380, partial = 41, none (stable or progressive disease) = 58.) (One-way ANOVA with multiple comparison tests, * *p* < 0.05, *** *p* < 0.001.) KM survival curves show high levels of miR-21 significantly associated with (**D**) decreased overall survival and (**E**) reduced disease-free interval. (Both log-rank (Mantel–Cox) test, * *p* < 0.05, ** *p* < 0.01.) Data analysis for B to E was performed using UCSC Xena based on PRAD cohort in TCGA database. All boxplots show mean and Tukey whiskers.

**Table 1 cancers-15-01291-t001:** Functional enrichment analysis of miR-21 in prostate cancer. Table shows the significant association of miR-21 target genes with Gene Set descriptions related to prostate cancer. Analysis performed using clusterProfiler in CancerMIRNome.

Gene Set	Gene Set ID	Description	Count/Total	Adjusted *p*-Value ^1^	Gene Symbol
**KEGG**	hsa05206	MicroRNAs in cancer	35/612	1.32 × 10^−7^	*CDC25A*; *BCL2*; *SPRY2*; *TIMP3*; *RECK*; *E2F2*; *PTEN*; *E2F1*; *MARCKS*; *TPM1*; *CDK6*; *PDCD4*; *SERPINB5*; *BMPR2*; *MYC*; *ERBB2*; *HNRNPK*; *TP63*; *EGFR*; *NFKB1*; *VEGFA*; *MDM4*; *TGFB2*; *PIK3R1*; *MMP9*; *BRCA1*; *PRKCE*; *APC*; *CCNG1*; *STAT3*; *DICER1*; *E2F3*; *ABCB1*; *BMI1*; *SOCS1*
	hsa05215	Prostate cancer	16/612	8.95 × 10^−6^	*BCL2*; *E2F2*; *PTEN*; *E2F1*; *ERBB2*; *EGFR*; *PLAT*; *NFKB1*; *RB1*; *PDGFD*; *PIK3R1*; *MMP9*; *AKT2*; *IGF1R*; *E2F3*; *FOXO1*
**Disease Ontology**	DOID:10283	prostate cancer	41/612	2.03 × 10^−5^	*BCL2*; *SPRY2*; *PTEN*; *HIPK3*; *FAS*; *BMPR2*; *MYC*; *ERBB2*; *TOPORS*; *MSH2*; *EGFR*; *ICAM1*; *SP1*; *SMARCA4*; *NFKB1*; *SOD3*; *SMAD7*; *MMP2*; *VEGFA*; *TGFB1*; *RB1*; *PDGFD*; *MUC1*; *PIK3R1*; *RPS6KA3*; *MMP9*; *BRCA1*; *PTK2*; *SKP2*; *PBX1*; *WNT5A*; *MAP3K1*; *PURA*; *HIF1A*; *CXCL10*; *IGF1R*; *SET*; *KLK2*; *CEBPB*; *BMI1*; *CASP8*
	DOID:10286	prostate carcinoma	13/612	9.53 × 10^−3^	*BCL2*; *PTEN*; *MYC*; *ERBB2*; *MSH2*; *EGFR*; *NFKB1*; *MMP2*; *VEGFA*; *PDGFD*; *PTK2*; *IGF1R*; *CASP8*
**DisGeNET**	umls:C0936223	Metastatic Prostate Carcinoma	24/612	5.11 × 10^−6^	*JAG1*; *PTEN*; *MYC*; *ERBB2*; *EGFR*; *IL1B*; *NFKB1*; *NTF3*; *DTX3L*; *MMP2*; *VEGFA*; *PARP1*; *TGFB1*; *ACAT1*; *SUZ12*; *MUC1*; *MMP9*; *PARP9*; *WNK1*; *TNFRSF11B*; *SATB1*; *WWP1*; *HIF1A*; *CLU*
	umls:C0007112	Adenocarcinoma of prostate	16/612	4.22 × 10^−4^	*BCL2*; *PTEN*; *ERBB2*; *EGFR*; *PPARA*; *PLPP1*; *VEGFA*; *TGFB1*; *RB1*; *MMP9*; *PRKCE*; *MIB1*; *TLR4*; *OLR1*; *STAT3*; *HIF1A*
	umls:C1328504	Hormone refractory prostate cancer	11/612	4.44 × 10^−4^	*BCL2*; *PTEN*; *ERBB2*; *EGFR*; *PARP1*; *TGFB1*; *APC*; *STAT3*; *CLU*; *HMGB1*; *CASP8*
	umls:C1654637	androgen independent prostate cancer	15/612	9.38 × 10^−4^	*BCL2*; *PTEN*; *ERBB2*; *MEF2C*; *EGFR*; *RASGRP3*; *MMP9*; *PBX1*; *AGO2*; *AKT2*; *FOXO3*; *HIF1A*; *CLU*; *COX2*; *ABCB1*

KEGG = Kyoto Encyclopedia of Genes and Genomes. ^1^ Adjusted *p*-value for multiple hypothesis correction used Benjamini and Hochberg procedure.

**Table 2 cancers-15-01291-t002:** Functional enrichment analysis of miR-21 related to hypoxia. Table shows the significant association of miR-21 target genes with Gene Set descriptions related to hypoxia. Analysis performed using clusterProfiler in CancerMIRNome.

Gene Set	Gene Set ID	Description	Count/ Total	Adjusted *p*-Value ^1^	Gene Symbol
**KEGG**	hsa04066	HIF-1 signaling pathway	15/612	8.10 × 10^−5^	*BCL2*; *PDHA2*; *ERBB2*; *EGFR*; *NFKB1*; *VEGFA*; *MKNK2*; *PIK3R1*; *TLR4*; *AKT2*; *STAT3*; *VHL*; *HIF1A*; *IGF1R*; *EGLN1*
**Gene Ontology-** **Biological Process**	GO:0001666	response to hypoxia	29/612	2.31 × 10^−4^	*BCL2*; *REST*; *TGFBR2*; *PTEN*; *E2F1*; *APAF1*; *MYC*; *TGFBR3*; *ICAM1*; *PLAT*; *PPARA*; *SOD3*; *MMP2*; *VEGFA*; *TGFB1*; *MDM4*; *DDAH1*; *TGFB2*; *APOLD1*; *PRKCE*; *IRAK1*; *VHL*; *FOXO3*; *HIF1A*; *SIRT2*; *DNM1L*; *STUB1*; *EGLN1*; *PSMD9*
	GO:0071456	cellular response to hypoxia	17/612	4.70 × 10^−3^	*BCL2*; *PTEN*; *E2F1*; *MYC*; *ICAM1*; *VEGFA*; *MDM4*; *DDAH1*; *PRKCE*; *IRAK1*; *VHL*; *FOXO3*; *HIF1A*; *SIRT2*; *STUB1*; *EGLN1*; *PSMD9*
	GO:0070482	response to oxygen levels	32/612	9.66 × 10^−5^	*BCL2*; *REST*; *TGFBR2*; *PTEN*; *E2F1*; *APAF1*; *FAS*; *MYC*; *TGFBR3*; *ICAM1*; *PLAT*; *PPARA*; *SOD3*; *MMP2*; *VEGFA*; *TGFB1*; *MDM4*; *DDAH1*; *TGFB2*; *APOLD1*; *PRKCE*; *IRAK1*; *VHL*; *FOXO3*; *HIF1A*; *SIRT2*; *DNM1L*; *STUB1*; *EGLN1*; *PSMD9*; *OXTR*; *FOXO1*
	GO:0036293	response to decreased oxygen levels	30/612	1.66 × 10^−4^	*BCL2*; *REST*; *TGFBR2*; *PTEN*; *E2F1*; *APAF1*; *MYC*; *TGFBR3*; *ICAM1*; *PLAT*; *PPARA*; *SOD3*; *MMP2*; *VEGFA*; *TGFB1*; *MDM4*; *DDAH1*; *TGFB2*; *APOLD1*; *PRKCE*; *IRAK1*; *VHL*; *FOXO3*; *HIF1A*; *SIRT2*; *DNM1L*; *STUB1*; *EGLN1*; *PSMD9*; *OXTR*
	GO:0071453	cellular response to oxygen levels	20/612	1.37 × 10^−3^	*BCL2*; *PTEN*; *E2F1*; *FAS*; *MYC*; *ICAM1*; *VEGFA*; *MDM4*; *DDAH1*; *PRKCE*; *IRAK1*; *VHL*; *FOXO3*; *HIF1A*; *SIRT2*; *DNM1L*; *STUB1*; *EGLN1*; *PSMD9*; *FOXO1*
	GO:0036294	cellular response to decreased oxygen levels	18/612	3.27 × 10^−3^	*BCL2*; *PTEN*; *E2F1*; *MYC*; *ICAM1*; *VEGFA*; *MDM4*; *DDAH1*; *PRKCE*; *IRAK1*; *VHL*; *FOXO3*; *HIF1A*; *SIRT2*; *DNM1L*; *STUB1*; *EGLN1*; *PSMD9*
	GO:0034599	cellular response to oxidative stress	21/612	7.61 × 10^−3^	*BCL2*; *REST*; *TPM1*; *RHOB*; *EIF2S1*; *PPIF*; *EGFR*; *TNFAIP3*; *SOD3*; *MMP2*; *PARP1*; *PDGFD*; *PKD2*; *PLEKHA1*; *MMP9*; *TLR4*; *FOXO3*; *HIF1A*; *SIRT2*; *PCGF2*; *FOXO1*
	GO:0006979	response to oxidative stress	26/612	1.90 × 10^−2^	*BCL2*; *REST*; *TPM1*; *SESN1*; *RHOB*; *EIF2S1*; *PPIF*; *EGFR*; *TNFAIP3*; *SP1*; *SOD3*; *MMP2*; *PARP1*; *PDGFD*; *MYEF2*; *PKD2*; *PLEKHA1*; *MMP9*; *TLR4*; *FOXO3*; *HIF1A*; *SIRT2*; *CCR7*; *EGLN1*; *PCGF2*; *FOXO1*

KEGG = Kyoto Encyclopedia of Genes and Genomes. ^1^ Adjusted *p*-value for multiple hypothesis correction using Benjamini and Hochberg procedure.

## Data Availability

The genotypic and phenotypic data for Prostate adenocarcinoma (PRAD) cohort are available at The Cancer Genome Atlas (TCGA) portal (http://portal.gdc.cancer.gov/projects, accessed on 11 January 2022). Serum miR-182 expression data are available at Gene Expression Omnibus (GEO) portal (http://www.ncbi.nlm.nih.gov/geo/, accessed on 13 May 2022); datasets GSE112264, GSE139031, GSE113486 and GSE134266. Analysis tools are listed in the Methods section and the other datasets analysed in the present study are available from the published papers that have been cited in this manuscript.

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
