# Peer review of "MiR-21 Is Induced by Hypoxia and Down-Regulates RHOB in Prostate Cancer"

_cancers, 2023, doi:10.3390/cancers15041291_

Round 1

Reviewer 1 Report

Summary

This study sought to investigate the role of ‘oncomiR’, miR-21 in prostate cancer (PCa). It’s altered expression has been previously shown to drive progression in a range of cancers, including PCa. Hypoxia is a known common characteristic in the PCa tumour microenvironment and is known to promote aberrant miRNA expression. Despite these links little is known about the impact of hypoxia on miR-21 expression and how this might contribute to PCa progression, which is what the authors sought to investigate. 

The authors confirm that miR-21 is upregulated in PCa and that in patient samples increased expression correlates with a range of clinical parameters. Observing that that hypoxia drives this change in miR-21 expression. They establish links between miR-21 and RHOB observing an an inverse correlation. This relationship was explored through the overexpression of miR-21 in normal prostate cells, where a decrease in RHOB expression was noted along with an induced PCa phenotype of increased migration and colony formation. Lastly, they highlight miR-21’s potential as a future biomarker to predict treatment response. 

Overall, the paper is well written and addresses a new area in PCa research. The authors must be commended for their use of multiple in vivo and in vitro samples, thus highlighting potential clinical translation. 

There are however a few minor areas of clarification required before the paper can be published. 

Major

·      In Lines 63-64  the authors state “highlights miR-21 is abnormally expressed in prostate cancer”. Can they expand this to provide detail on whether these studies show increased or decreased expression, providing specific references linking to the original research that supports this statement. 

·      Why did the authors choose to investigate RHOB over PTEN?  Table 2 had multiple targets, where RHOB was only associated with oxidative stress rather than hypoxia. They also investigated PTEN which appeared in table 2 associated with hypoxia, was also regulated by miR-21(Figure 3b) and is known to linked to PCa. 

·      Methods.  Bioinformatic analysis was employed extensively throughout however the associated methods section was brief. There is a need to clarify some bioinformatics methods and add more detail, potentially in the supplementary section. (1) In Figure 1 the use of UCSC XENA TCGA, was the data normalised and did it undergo quality control. (2) Functional enrichment analysis – more details needed on how this was done. What samples / database did they use to produce this figure? Was it PCa samples? Lines 259-262. (3) Figure 5F, what methods were employed to generate this figure. Also, can the authors provide better explanation of the numbers provided and how it shows the two miRNAs predict response. It appears that Gleason score alone is a better predicator at 62%. This section and their point was not clear to the reader. What is the advantage of combining the two miRNAs over using Gleason score alone? (4) Multiple hypothesis testing was noted at various stages but it is not clear if this was calculated by the employed software or done manually, if it was the later associated details should be included,

·      Results 3.4 – Did the authors confirm significant overexpression of miR-21 in their RWPE-1 this would need to be included in the supplementary.

Minor 

·      Lines 55 – 59  - Sentence is too long, needs restructured. 

·      Figure 2B bar lines appear thicker compared to others this needs corrected. Also this is the same for Figure 3B vs 3C. 

·      Lines 278-280 – Hypoxia presence confirmed in both models? The supplementary figure 2 needs to be updated to outline what blots refer to LNCaP cells and those related to spheroids. 

·      Supplementary figure 5 B and C, legend needs updated to outline that blue line(?) corresponds to low RHOB expression. 

Author Response

Thank you for the review of the above paper. We appreciate the time you have taken and welcome the insightful comments, which have helped to improve the paper.

We have itemised our responses below. Where appropriate, we have amended our manuscript accordingly and tracked the changes. Where line numbers are indicated, this refer to the CLEAN version of the manuscript.

We trust you find these revisions satisfactory.

This study sought to investigate the role of ‘oncomiR’, miR-21 in prostate cancer (PCa). It’s altered expression has been previously shown to drive progression in a range of cancers, including PCa. Hypoxia is a known common characteristic in the PCa tumour microenvironment and is known to promote aberrant miRNA expression. Despite these links little is known about the impact of hypoxia on miR-21 expression and how this might contribute to PCa progression, which is what the authors sought to investigate.

The authors confirm that miR-21 is upregulated in PCa and that in patient samples increased expression correlates with a range of clinical parameters. Observing that that hypoxia drives this change in miR-21 expression. They establish links between miR-21 and RHOB observing an an inverse correlation. This relationship was explored through the overexpression of miR-21 in normal prostate cells, where a decrease in RHOB expression was noted along with an induced PCa phenotype of increased migration and colony formation. Lastly, they highlight miR-21’s potential as a future biomarker to predict treatment response.

Overall, the paper is well written and addresses a new area in PCa research. The authors must be commended for their use of multiple in vivo and in vitro samples, thus highlighting potential clinical translation.

There are however a few minor areas of clarification required before the paper can be published.

  We are pleased the reviewer finds the work to be well-written, novel and suitably conducted. We endeavour to address the concerns below.

Major

In Lines 63-64  the authors state “highlights miR-21 is abnormally expressed in prostate cancer”. Can they expand this to provide detail on whether these studies show increased or decreased expression, providing specific references linking to the original research that supports this statement.

Almost all studies show miR-21 is up-regulated in prostate cancer, compared to normal tissue, so we have reworded the sentence to indicated that it is ‘over-expressed’ in prostate cancer (line 67). A citation has been added to support this statement [Ref. 17], and the over-expression of miR-21 in prostate cancer is further emphasised in the Discussion, citing appropriate references [Refs 90,91].

Why did the authors choose to investigate RHOB over PTEN?  Table 2 had multiple targets, where RHOB was only associated with oxidative stress rather than hypoxia. They also investigated PTEN which appeared in table 2 associated with hypoxia, was also regulated by miR-21(Figure 3b) and is known to linked to PCa.

PTEN had already been investigated and validated as a target of miR-21, both in a prostate cancer setting and in relation to hypoxia. We therefore decided there was no novelty in simply showing that relationship again. While we did use it in some experiments as a ‘positive control’ (ie to confirm our miR-21 over-expression was working), we wanted to identify a novel target, which no-one had investigated in relation to hypoxia and prostate cancer. That was why we chose RHOB instead.

Methods.  Bioinformatic analysis was employed extensively throughout however the associated methods section was brief. There is a need to clarify some bioinformatics methods and add more detail, potentially in the supplementary section.

The computational tools used are all published, validated methods recommended by National Cancer Institute for TCGA data analysis (https://www.cancer.gov/about-nci/organization/ccg/research/structural-genomics/tcga/using-tcga/tools). Each one uses bespoke algorithms and bioinformatic pipelines to download, process, normalize TCGA data, for subsequent interrogation by researchers. Robust statistical analyses are built into the interface for researchers, but data can also be downloaded for manual stats testing if preferred. It would be prohibitively long to include detail on each in this paper, but substantial detail on the design of each tool is provided in the citations given.

 (1) In Figure 1 the use of UCSC XENA TCGA, was the data normalised and did it undergo quality control.

Yes, as indicated at the UCSC website, all data is normalised and subject to quality control as part of the published methodology utilized to pre-process the cancer omics data from TCGA for analysis by users of the Xenabrowser tool. We have added text to confirm the data is pre-processed and normalized for analysis (Section 2.7)

(2) Functional enrichment analysis – more details needed on how this was done. What samples / database did they use to produce this figure? Was it PCa samples? Lines 259-262.

Yes, this was based on TCGA-PRAD data (ie prostate cancer samples). The CancerMIRNome website utilizes an integrated Bioconductor package called clusterProfiler to perform functional enrichment analysis on selected TCGA datasets. We chose PRAD dataset and applied clusterProfiler analyses using the instructions provided. clusterProfiler is a simple-to-use ontology-based tool for gene classification and enrichment analyses. As explained in the citation (Ref 52), it is based on statistical analysis of biomedical gene set and KEGG databases, thereby allowing biological theme comparison among gene clusters. We have now added text to methods to clarify this (Section 2.7).

(3) Figure 5F, what methods were employed to generate this figure. Also, can the authors provide better explanation of the numbers provided and how it shows the two miRNAs predict response. It appears that Gleason score alone is a better predicator at 62%. This section and their point was not clear to the reader. What is the advantage of combining the two miRNAs over using Gleason score alone?

Upon re-reading we do acknowledge Figure 5F is potentially confusing for the reader. A decision Tree is a non-parametric method that defines thresholds for each variable to help divide the dataset into categories. This one was intended to demonstrate that the combination of the three variables were better at identifying those patients that experienced remission, compared to using Gleason grade alone (ie the combination would predict remissions that the ‘Gleason only’ arm had missed, which is the node that the Reviewer is referring to, at 62% accuracy). The addition of miR-21 and miR-210 captures patients that the branch GS >=9 originally missed, thus improving the specificity. However, this is only one representative decision tree of how measurements might be combined, rather than a definitive result. Further iterations might change order or threshold of variable to best divide the cohort. Since it was merely an illustrative example and not crucial to the paper conclusions and in the interest of brevity, we have removed the decision tree from Figure 5 altogether.

That said, we still propose that multivariate analysis is likely to be better at accurate prediction of patient outcome, and that miR-21 and miR-210 measurements are useful variables to include in such prognostic analysis. We have therefore kept the random forest data, which is generated from multiple decision tree iterations (ie if one decision tree is good, then many should be better) to illustrate this potential. We have moved it from 5G to Supplementary Table S1 as this preliminary model would need to be validated with independent training and test datasets, and have included text to this effect. We have also amended text in manuscript throughout which previously discussed Fig 5F and 5G.

(4) Multiple hypothesis testing was noted at various stages but it is not clear if this was calculated by the employed software or done manually, if it was the later associated details should be included,

This was performed automatically by the bioinformatic tools used, meaning we had no extra manual adjustments for multiple hypothesis testing to apply afterwards. Details are provided in the accompanying citations for each software tool. For example, to prevent high false discovery rate (FDR) in multiple testing, clusterProfiler and Firebrowse estimated q-values to control for FDR.

  • Results 3.4 – Did the authors confirm significant overexpression of miR-21 in their RWPE-1 this would need to be included in the supplementary.

Yes, we routinely confirmed the over-expression of miR-21 following transfection experiments. This data has now been added to Figure 4 (panel A).

Minor

  • Lines 55 – 59  - Sentence is too long, needs restructured.

These lines has now been rewritten (lines 55-59).

  • Figure 2B bar lines appear thicker compared to others this needs corrected. Also this is the same for Figure 3B vs 3C.

The graphs have been reformatted for consistency

  • Lines 278-280 – Hypoxia presence confirmed in both models? The supplementary figure 2 needs to be updated to outline what blots refer to LNCaP cells and those related to spheroids.

The figure has been amended to show which blots refer to cells and spheroids

  • Supplementary figure 5 B and C, legend needs updated to outline that blue line(?) corresponds to low RHOB expression.

This detail has been added to the legend, indicating that red lines represent high RHOB expression and blue lines represent low RHOB expression.

Reviewer 2 Report

This paper explores role of MiR-21 in prostate cancer. Authors show that MiR-21 is upregulated by hypoxia and may contribute to tumor progression by targeting RHOB. Methodology is sound and experiments performed stringently. Paper is well written. I have only a few minor comments:

In the abstract the authors propose 'MiR-21 as a clinical useful diagnostic and prognostic biomarker of hypoxia and urological malignancies cancers including prostate cancer'. Sentence needs to be rephrased and maybe modified to embrace only prostate cancer, since this is what's approached in the paper.

One big problem with biomarkers in prostate cancer is the heterogeneity of the disease, which has also been documented with regard to MiR's (see Zedan et al. PlosOne 2017; Jun 19; 12(6): e0179113). Please reflect on this in the context of the present data.

Author Response

Thank you for the review of the above paper. We appreciate the time you have taken and welcome the insightful comments, which have helped to improve the paper.

We have itemised our responses below. Where appropriate, we have amended our manuscript accordingly and tracked the changes. Where line numbers are indicated, this refer to the CLEAN version of the manuscript.

We trust you find these revisions satisfactory.

This paper explores role of MiR-21 in prostate cancer. Authors show that MiR-21 is upregulated by hypoxia and may contribute to tumor progression by targeting RHOB. Methodology is sound and experiments performed stringently. Paper is well written. I have only a few minor comments:

We are pleased the reviewer finds the work to be well-written and suitably conducted.

In the abstract the authors propose 'MiR-21 as a clinical useful diagnostic and prognostic biomarker of hypoxia and urological malignancies cancers including prostate cancer'. Sentence needs to be rephrased and maybe modified to embrace only prostate cancer, since this is what's approached in the paper.

We accept this phrase over-reaches on what is presented in the paper, so we have amended accordingly (Line 37)

One big problem with biomarkers in prostate cancer is the heterogeneity of the disease, which has also been documented with regard to MiR's (see Zedan et al. PlosOne 2017; Jun 19; 12(6): e0179113). Please reflect on this in the context of the present data.

The reviewer raises an important point as tumor heterogeneity is a major issue in prostate cancer (and other malignancies), both in the interpretation of data and in the extrapolation to clinical usefulness. With that in mind, we have added sentences to acknowledge this issue and to highlight the challenges (Lines 564-575) supported by the citation suggested by reviewer and two others (New refs 100-102). We additionally explain that expression of RHOB and/or miR-21 may well be dependent on certain cell-types within the tumor. We propose the single-cell analysis and/or focus on disease sub-types as prudent approaches to unravel their combined biological function of RHOB/miR-21.

Reviewer 3 Report

In this manuscript, Angel et al. investigate the role of miR-21 in prostate cancer and its related mechanisms involved in tumor hypoxia. Of interest, the authors link the hypoxia and prostate cancer by miR-21, and highlight the importance of miR-21/RHOB axis in prostate cancer progression. The findings are interesting, and the experiments are convincing, except for the link between miR-21 and RHOB. The following points must be adequately addressed before the paper can be recommended for publication.

Concern points:

1.    The author should clarify whether the miR-21/RHOB axis is direct or not and whether RHOB is mandatory for miR-21-mediated biological functions.

2.    The author should check the effect of miR-21/RHOB axis on mice model(s).

3.    How about the correlation between miR-21 and RHOB in biochemical recurrence patient samples?

4.    How about the RHOB and PTEN protein expression levels upon miR-21 OE/KD?

Author Response

Thank you for the review of the above paper. We appreciate the time you have taken and welcome the insightful comments, which have helped to improve the paper.

We have itemised our responses below. Where appropriate, we have amended our manuscript accordingly and tracked the changes. Where line numbers are indicated, this refer to the CLEAN version of the manuscript.

We trust you find these revisions satisfactory.

  1. The author should clarify whether the miR-21/RHOB axis is direct or not and whether RHOB is mandatory for miR-21-mediated biological functions.

We already know that RHOB is a direct target of miR-21 from previous studies, which we have cited in the paper (refs 61-63), and from the evidence available in miRTarBase database of experimentally linked miR-21 targets.

We were primarily interested in the genomic link between miR-21 and RHOB in prostate cancer, so our aim here was to gather evidence to corroborate data from those previous studies setting and compare with the genomic data in TCGA PRAD repository, which we have done.

However, we do accept that further work (e.g. analyzing RhoB at protein level) could help definitively validate RHOB as a direct target in prostate cancer cells and have now added text to acknowledge that (Lines 508-511). We have now proposed a changed title also which more accurately describes the findings presented (Line 2). In line with this, we have also changed the text throughout the manuscript to be less definitive about stating RHOB as a target of miR-21. Instead, we now state simply that miR-21 down-regulates RHOB mRNA expression (ie an inverse correlation between miR-21 and RHOB exists at the genomic level)(Section 3.3). Likewise, throughout manuscript, to make this distinction clear we now utilise capitals and italics (RHOB) to show we are referring to the gene rather than the protein (RhoB). This convention now applies to all genes and proteins mentioned in the manuscript (See also point 4 below).

Whether RHOB is specifically needed to exert miR-21 function is a more complicated query, which extends to the entire area of miRNA-target interactions. In our opinion, miR-21 has such a broad range of targets and functions, that it is difficult to see how any one of these would be considered ‘mandatory’. Indeed the promiscuity of miRNAs in having many targets is one of the factors that makes them such powerful regulators as it provides scope for control of parallel and overlapping signalling pathways, even if one target within those pathways becomes compromised. There are also multiple feedback and feedforward mechanisms, which mean miRNAs can alter function through a variety of direct and indirect interactions. So, while RHOB may be a target of miR-21, there is nothing to suggest it is a ‘bottleneck’ through which miR-21 must act. As we have alluded to in the paper, it is the holistic effect of miR-21 that must be considered. Naturally, this hypothesis would need tested with specific studies, but that would be a study in its own right (see point 2 below also).

  1. The author should check the effect of miR-21/RHOB axis on mice model(s).

The in vivo data presented in this paper utilised samples collected from mice as part of a previous study examining tumour hypoxia in a xenograft model. As such, the focus of the paper was on profiling expression of interesting miRNAs and mRNA targets in the cellular response to hypoxia, which we could then compare with related in vitro and in silico data. We therefore view this current manuscript to be ‘proof-of-principle’ providing evidence that miR-21 and RHOB are linked and worthy of further investigation. However, focused experiments on the effect of miR-21/RHOB axis in vivo (such as over-expression/knockdown of either molecule in mice) was beyond the scope of this study and would constitute a separate paper in its own right, using a new, specifically-designed mouse experiment to conduct the study.

That said, we do agree with the reviewer that this would be a prudent approach in any future work looking to unravel the biological mechanisms linked to the miR-21/RHOB axis, especially in the context of hypoxia. To acknowledge this we have added more sentences to suggest this as further work, citing papers which have looked at hypoxia-related miR-21 function in mouse experiments as an example of how this would be performed. (lines 534-540).

  1. How about the correlation between miR-21 and RHOB in biochemical recurrence patient samples?

On the reviewer’s suggestion, we have performed this analysis. In patients with recurrence (n = 61) and in patients without recurrence (n = 406), there was still a significantly inverse correlation between miR-21 and RHOB in both groups. These graphs have been added to Supplementary Figure S5 (B & C).

  1. How about the RHOB and PTEN protein expression levels upon miR-21 OE/KD?

As we mention in Point 1 above, we already knew RHOB was a validated target of miR-21 and data we present would corroborate that. Also, we were primarily interested in the miRNA:mRNA data since we were comparing it with genomic data from TCGA.

However, we do accept that for future work on miR-21/RHOB it would be sensible to look at RhoB protein levels too and we have inserted a sentence to acknowledge that (Lines 508-511). With that in mind, we have amended the title of Figure 3 to acknowledge this data presents RHOB as a likely target, rather than a definitive one. Similarly, we have also changed the text throughout to be less definitive about stating RHOB as a target of miR-21. Instead, we now state simply that miR-21 down-regulates RHOB mRNA expression. Likewise, throughout manuscript, to make this clear we now utilise capitals and italics (RHOB) to show we are referring to the gene rather than the protein (RhoB) (See also point 1 above).

Round 2

Reviewer 3 Report

The authors have addressed my some concerns, however, some validation must be performed in their hand.

The author highlight RHOB is a direct target of miR-21 from previous studies, do other group use the LNCaP cell line? If not, please validate this regulation.

Author Response

Thank you again for your extra review, which has again helped to improve the paper.

We have provided our responses below and amended our manuscript accordingly. The line numbers refer to the CLEAN version of the manuscript.

We trust you find these revisions satisfactory.

The author highlight RHOB is a direct target of miR-21 from previous studies, do other group use the LNCaP cell line? If not, please validate this regulation.

Since the other studies we cite did not report using LNCaP cells, we agreed that it would be useful to provide further experimental proof that that miR-21 targets RHOB in LNCaP cells.

Therefore, we performed Western blotting to show that over-expressing miR-21 in LNCaP cells does indeed reduce the protein levels of RhoB, which provides the extra validation that it is a direct target.

We confirmed this in two separate biological replicates and show one of the representative results in the revised Figure 3 (Panel B). Text relevant to this new addition is added in Methods (Lines 161-2), Results (Lines 344-5), Discussion (Lines 507-14).